# Antibodies response in symptomatic and asymptomatic SARS-CoV-2 infected persons in Thailand

Chanida Ruchisrisarod[1]*, Phanni Wanthong[1], Yutthana Joyjinda[1], Saowalak Bunprakob[1], Pasin Hemachudha[1], Anek Mungaomklang[2], Thirawat Supharatpariyakorn[1], Thiravat Hemachudha[1], Abhinbhen Saraya Wasontiwong[1]

1 Thai Red Cross Emerging Infectious Diseases Health Science Centre, World Health Organization Collaborating Centre for Research and Training on Viral Zoonoses, King Chulalongkorn Memorial Hospital, Faculty of Medicine, Chulalongkorn University, Bangkok, Thailand, 2 Department of Disease Control, Ministry of Public Health, Institute for Urban Disease Control and Prevention, Bangkok, Thailand

* jeabchanida@gmail.com

**Data Availability Statement:** All relevant data are within the manuscript and its Supporting Information files.

## Abstract

Antibody assays of IgM, IgG and surrogate isotype independent virus neutralizing antibody (sVNT) targeting receptor binding domain of Severe Acute Respiratory Syndrome Coronavirus 2 (SARS-CoV-2) were employed in 97 real-time Reverse Transcription Polymerase Chain Reaction (RT-PCR) confirmed Coronavirus Disease 2019 (COVID-19) patients with varying severity admitted to King Chulalongkorn Memorial Hospital. Concordance rate was 100% regardless of severity, onset of symptoms and magnitude of viral load. Per available samples, antibodies appeared on the same day of symptom onset in one patient; one day after in 18 patients and two days after in 19 patients. In two patients, antibodies appeared as early as 4 days after infection (exposure). IgM and IgG were evident in all patients' first assay (within two days of admission). sVNT was also evident within two days of admission in all but 3 patients. IgM usually remained positive during the entire course of hospital stay, where the longest in this study was 32 days. Antibody assays were also applied to samples collected at a State Quarantine premise from 77 asymptomatic Thais returning from Sudan in October. Virus was detected by real-time RT-PCR in 15 cases (day 0 = 6, day 3 = 4, day 5 = 4 and day 9 = 1). Twenty-nine (including 11 RT-PCR positive cases) were antibody positive on day 0, while 4 PCR positive with antibody negative on day 0 became antibody positive on day 14. Evaluation on antibody response at days 7 or 10 is needed to help build a case to shorten length of quarantine among negative cases.

## Introduction

As the Coronavirus disease 2019(COVID-19) pandemic progresses into the latter half of 2020, research into the disease's pathogenesis has elucidated the importance of the host's immunological and adaptive response in predicting the severity and clinical outcomes of the disease. Our understanding or prediction of the specific innate and adaptive immune response to

**Funding:** The authors received no specific funding for this work.

**Competing interests:** The authors have declared that no competing interests exist.

Severe Acute Respiratory Syndrome Coronavirus 2 (SARS-CoV-2) remains limited. The World Health Organization reported that COVID-19 patients with mild symptoms or asymptomatic patients had low or undetectable levels of neutralizing antibodies (NAbs), and the converse is true [1], however, such results may be conflicting depending on the different techniques used. Despite its cost and need for specialized equipment and reagents, the current gold standard for detecting SARS-CoV-2 is molecular testing by real-time Reverse Transcriptase Polymerase Chain Reaction (RT-PCR) which detects genetic material of the virus from respiratory samples. This determines whether a person is currently infected with and shedding the virus. Sensitivity of PCR can be compromised by inadequacy of specimens, swab techniques, and inconsistent PCR results requiring repeated testing not only in symptomatic or patients under investigation (PUIs), but also in asymptomatic persons under quarantine.

On the other hand, serological testing detects antibodies against the virus. Although traditionally, serological tests are not indicative of viruses' transmission potential, an accurate and efficient serological test is necessitated to reduce cost, time, and increase test availability to help contain the pandemic. Further, serological tests can be used for retrospective contact tracing, case fatality determinations, rapid and inexpensive diagnosis of asymptomatic infection, seroprevalence assessments, humoral immunity assessment for use in screening for potential convalescent plasma donors [2] and monitoring immune response in vaccine candidates [3, 4]. Serological tests for SARS-CoV-2 antibody testing are essential for detecting current or past infections. While a wide range of commercial lateral flow tests to detect SARS-CoV-2 antigen is available, it is important to note that most of them require validation when compared with serological tests such as ELISA, which is the gold standard for detecting antibodies [5]. NAbs, produced naturally as part of the hosts' immune response, specifically detects viral surface epitopes that facilitate virus entry into a host cell [6, 7]. These epitopes are predominantly located in the receptor binding domain (RBD) of the spike protein in SARS-CoV-2 [8]. The gold standard for detecting NAbs is the virus neutralization test which requires BSL-3 laboratory. In order to test for NAbs, a more convenient and inexpensive assay was recently developed utilizing *in vitro* interactions between the RBD and the angiotensin converting enzyme 2 (ACE2) receptor proteins typically found on target host cells. NAbs competes with ACE-2 to bind RBD, thus the degree to which a patient's serum or plasma inhibits the RBD-ACE2 interaction correlates with the quantity of NAbs. This can be measured by calorimetrically labelling the purified proteins and is the basis of surrogate virus neutralization test (sVNT) which detects NAbs against SARS-CoV-2, where the RBD acts as the surrogate of the virus, and ACE2 as the surrogate of the susceptible cell [9]. sVNT as well as IgM, IgG assays based on this technique has been previously tested in two large cohorts in Asia, and is commercially available as cPass™, GenScript, USA. sVNT has been found to be a specific and sensitive assay for SARS-CoV-2 and proven to detect Nabs regardless of antibody isotypes In-house ELISA methods based on techniques described were developed to assay IgM, IgG and sVNT at the Thai Red Cross Emerging Infectious Diseases Health Science Centre (TRC-EID-HSC).

SARS-CoV-2 has been shown to mutate continually [10, www.gisaid.org, registered user], and re-infection has been reported [11]. SARS-CoV-2 genetic mutations can result in its newfound ability to escape the naturally developed immunity from the prior infection. Moreover, such mutations can affect the diagnostic accuracy of molecular detection such as real-time RT-PCR [10].

Shortening of quarantine period by 4–7 days from 14 days in asymptomatic PUI may be possible by using antibody testing to screen PUIs. Only antibody positive cases need to be further tested using real-time RT-PCR to determine their shedding status. This can reduce diagnostic costs and State Quarantine (SQ) burden.

This study aims to evaluate the sensitivity and specificity of serology assay and kinetics of antibody responses in 97 symptomatic patients in the first half of 2020. The study also evaluated its value as a screening tool among 77 asymptomatic persons in SQ who were returning from Sudan.

## Materials and methods

### Patients selection and samples collection

This study used leftover clinical specimens from 97 confirmed COVID-19 patients with varying severity grades which start on June 2020 and End on June 2021, admitted to King Chulalongkorn Memorial Hospital (KCMH) and 77 asymptomatic quarantined cases from Sudan in October 2020 with confirmed COVID-19 by Real-time reverse transcription polymerase chain reaction (rRT-PCR). All patients were tested for SARS-CoV-2 infection if they met the PUI (Patient Under Investigation) criteria which included a history of exposure to patients with COVID-19 or a history of travel to COVID-affected areas within the past 2 weeks. Two hundred blood samples from healthy donors at the Thai Red Cross Society National Blood Centre, prior to the pandemic outbreak, were included in this study as negative controls. Leftover blood from 300 persons infected by coronaviruses other than SARS-CoV-2, dengue, influenza and other respiratory viruses were also tested to confirm specificity. The study was approved by the Institutional Review Board at the Faculty of Medicine, Chulalongkorn University (IRB number 400/63) Written informed consent was obtained from all individual participants subjects before the study. Informed consent written informed consent was obtained from a legally authorized representative(s) for anonymized patient information to be published in this article.

Among patients admitted to KCMH, all fulfilled criteria at admission of having fever of greater than 37.5°C with or without cough, with sore throat or shortness of breath or pneumonia on chest radiograph, and with or without history of contact with a COVID-19 positive patient. Patients with bacterial infections were excluded from this study.

Among the 77 asymptomatic persons in SQ that returned from Sudan, nasopharyngeal swab samples were sequentially collected from all 77 upon admission into the SQ (day 0), and then on days 3, 5, 7, 9, 11 and 14. SARS-CoV-2 genetic testing was performed simultaneously by the Department of Medical Sciences and the Institute for Urban Disease Control and Prevention, Department of Disease Control, Ministry of Public Health, Thailand. Blood samples were collected on days 0 and 14.

Respiratory secretions from patients admitted to KCMH, such as nasal swab, nasopharyngeal swab, throat swab or sputum, and blood samples were collected and tested for SARS-CoV-2 using real-time RT-PCR at the TRC-EID-HSC as per previously published protocol [12]. Patients with PCR-confirmed COVID-19 were classified by disease severity as follows: mild–minimal symptom or evidence of upper respiratory tract infection; moderate–pneumonia without oxygen desaturation treated with antiviral agent(s) for 5 days; severe–pneumonia treated with oxygen support and antiviral agents for at least 10 days; critical–pneumonia requiring mechanical ventilation. These severity grades are analogous to the WHO blueprint grades 3, 4, and 5 respectively. These patients had serial Real-time reverse transcription polymerase chain reaction (rRT-PCR) and serology testing. Patients were discharged on the basis of clinical stability and negative Real-time reverse transcription polymerase chain reaction (rRT-PCR) results on 2 consecutive days.

Real-time reverse transcription polymerase chain reaction (rRT-PCR): Nasopharyngeal swabs and throat swab from all suspected COVID-19 patients underwent rRT-PCR testing for SARS-CoV-2 at the Thai Red Cross Emerging Infectious Diseases Health Science Centre of the Faculty of Medicine, Chulalongkorn University. Nucleic acid extraction was performed on all

samples using magDEA DX SV extraction kit according to the rRT-qPCR was performed using Seegene-Allplex 2019-nCov kit and Fosun novel corona kit, a CFX96 qPCR machine according to manufacturer's instruction. Briefly, reaction was heated to 50˚C for 20 minutes for reverse transcription, denatured in 95˚C for 15 minutes and then 45 cycles of amplification was carried in 94˚C for 15 seconds and 58˚C for 30 seconds. Fluorescence was measured using four fluorescence channels: FAM (E gene), Cal Red 610 (RdRp gene, HEX (internal control) for Seegene-Allplex 2019-nCov kit and FAM (ORF1ab gene), JOE (N gene), ROX (E gene), Cy5 (internal control) for Fosun novel corona kit.

SARS-CoV-2 antibody detection: ELISA in-house kit testing for IgG and IgM were developed based on protocol published by the Emerging Infectious Diseases Programme, Duke-NUS Medical School, Singapore (Tan et al. 2020) and recombinant proteins were purchased from GenScript. This in-house method has been compared with a commercial ELISA test kit for the detection of SARS-CoV-2 antibodies, demonstrating an efficient in-house test kit with acceptable values of sensitivity and specificity. ELISA test kits from GenScript were also used simultaneously. We prepared the ELISA plates for specifically detects human IgG antibodies directed against SARS-CoV-2 by diluting the antigen to a final concentration of 2 µg per ml of 2019-nCoV antigen. Then, we add fifty microliters (µl) per well to coat the 96-well microplate, which is stored at 4˚C overnight to test for ELISA IgG antibodies against SARS-CoV-2. Blocking buffer of 150 ml per well was incubated at room temperature for 120 minutes. Fifty µl of diluted serum specimens from patients at 1:200 was added to each well and incubated at 37˚C for 120 minutes. Afterward, the plates were washed five times with 300 µl of PBST per well. Following the wash, fifty µl of goat anti-human IgG conjugated to HRP, diluted at 1:10,000, was added to each well. The plates were then incubated at 37˚C for 60 minutes. Fifty µl of TMB substrate was added to each well and incubated in a dark room at room temperature and which the reaction was stopped by adding sulfuric acid (Stop solution). Optical density (OD) was detected at 450 nm. The captured IgM antibodies were prepared by diluting the antigen to a final concentration of 10µg/ml 2019-nCoV antigen Then, we add fifty microliters (µl) per well to coat the 96-well microplate, which is stored at 4˚C overnight for SARS-CoV-2 capture IgM antibodies against SARS-CoV-2. Blocking buffer of 150 ml per well was incubated at room temperature for 120 minutes. Fifty µl of diluted serum specimens from patients at 1:50 was added to each well and incubated for 120 minutes and incubated at 37˚C. Afterward, the plates were washed five times with 300 µl of PBST per well. Following the wash, fifty µl of HRP-RBD at 4µg per ml was added to each well and incubated at 37˚C for 60 minutes. Fifty µl of TMB substrate was added to each well and incubated in a dark room at room temperature and which the reaction was stoped by adding sulfuric acid (Stop solution). Optical density (OD) was detected at 450 nm.

Neutralizing antibodies (NAbs) tests: The sVNT kit, specifically the cPass™ assay from GenScript USA, following the manufacturer's instructions. This assay detects and measures circulating neutralizing antibodies against the receptor binding domain of severe acute respiratory syndrome coronavirus 2 (SARS-CoV-2). Serum specimens from patients diluted 1:9 ratio with sample diluent solution was mixed with horseradish peroxidase (HRP)–RBD was pre-incubated at 1:1 ratio to test for sVNT. after which it was added onto the ELISA plate pre-coated with human ACE2 100 µl of the mixture was then added to each of the 96 wells in the microplate and incubated at 37˚C for 15 minutes and the plates were washed three times with 300 µl of washing solution per well. Following the wash. TMB substrate at 100µl per well was added and incubated in a dark room at room temperature for 15 minutes, washed three times with 300 µl of washing solution per well. Following the wash, and the reaction was stopped using stop solution. OD was detected at 450nm, with positive control emitting highest intensity of light, with OD values of less than 0.3, while negative control emitted the lowest intensity with

OD value of more than 1, and the reagent did not change colour. The cutoff ratio for percentage of inhibition was at 20%.

In-house testing (IgG and IgM assays) and sVNT was validated against 97 blood samples of real-time RT-PCR confirmed COVID-19 patients, 200 negative controls and 300 relevant controls with other infections. OD value cut-off for IgM and IgG was 0.6, while percentage inhibition greater than 20% was considered positive. Results were in accordance with commercial GenScript test kits. In-house testing was then applied to 251 samples from these 97 patients as well as left over samples collected on days 0 and 14 from 77 asymptomatic persons in quarantine premise from Sudan. Detection of viruses in this latter group was done using two real-time RT-PCR test kits against RdRP and N genes in one, and E, RdRP and N in the other. Positive result from one or both kits were considered positive. In-house testing and sVNT ELISA of sera to detect for antibodies against 2019-nCoV protein (IgG) (Figs 1 and 2 and Table 1).

## Statistical analysis

The results are presented as mean ± standard deviation (SD) or median and interquartile range. Differences among groups were analysed using the One-way ANOVA. Statistical analyses were performed using RStudio Version 1.4.1564 on MacOS 10.15.7. Statistical significance was determined to be P-value = 0.006. A demographic table is needed to compare each group using SPSS version2.9. Statistical significance was determined to be P-value = 0.001.

## Results and discussion

Ninety-seven PCR-positive COVID-19 patients were included in this study which were representatives of the first half of the year 2020. Fifty-three (55%) were female. Age for 84 patients (86.6%) were known, with median age of 36 (range 6–76 years). Three patients (3.6%) were less than 20 years old, 53 patients (63.1%) were between ages 20 and 39, 24 patients (28.5%) were between ages 40 and 59, and 4 patients (4.8%) were between ages 60 and 76. Cycle-threshold (Ct) values from real-time RT-PCR assays on admission were available in all but one cases.

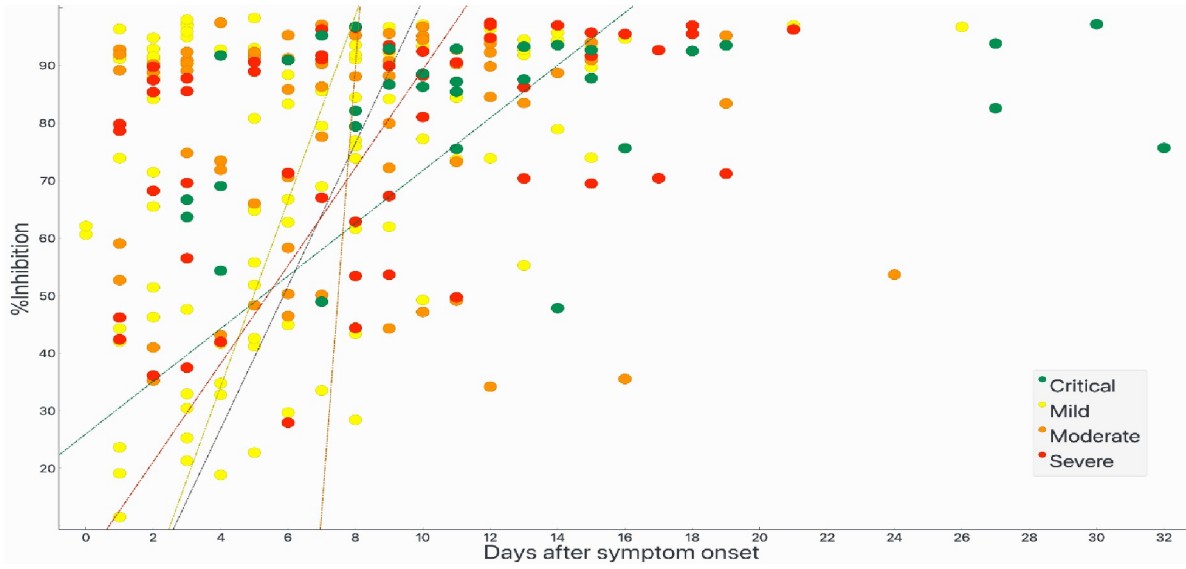

**Fig 1. Analyses of 97 PCR-positive COVID-19 patients against % inhibition of sVNT for the different disease severity categories: Mild or upper respiratory tract infection; moderate or pneumonia without oxygen desaturation; severe or pneumonia with desaturation, critical or pneumonia requiring mechanical ventilation.**

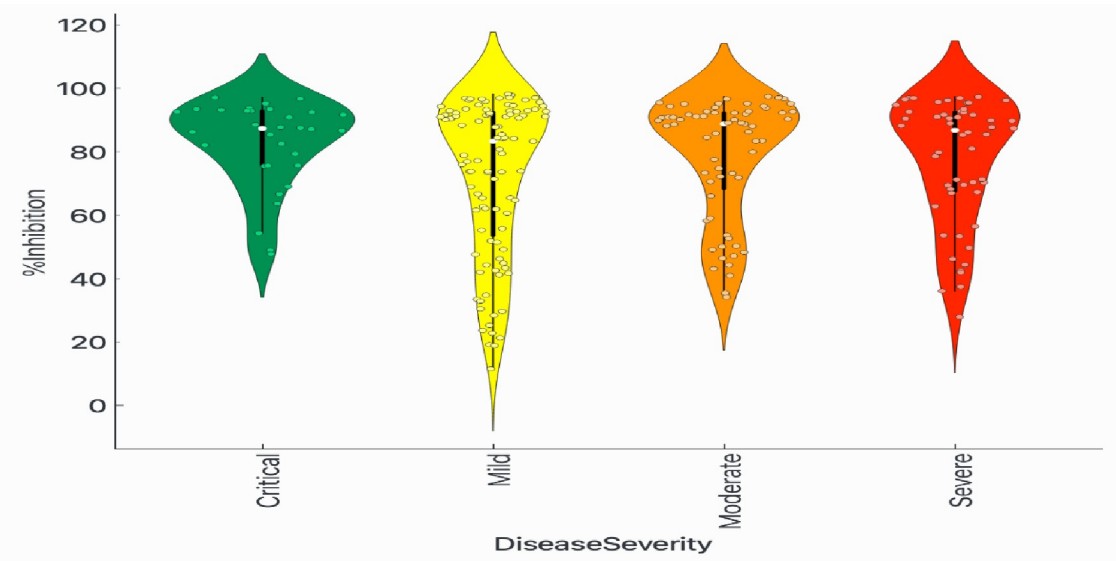

**Fig 2. Analyses of 97 PCR-positive COVID-19 patients were conducted against % inhibition of sVNT values for different disease severity categories: Mild or upper respiratory tract infection, moderate or pneumonia without oxygen desaturation, severe or pneumonia with desaturation, and critical or pneumonia requiring mechanical ventilation, using one-way ANOVA.**

Five of them (5.4%) had a Ct value of over 35 (considered as low viral loads). Percentage of patients with mild, moderate, severe, and critical were 53.6% (52/97), 22.7% (22/97), 13.4% (13/97), and 11.3% (11/97), respectively. Average Ct values in patients did not differ or pre-empt different disease severity. For example, average value of Ct in almost asymptomatic or mild symptoms was 22.16, while average value of Ct in critical patients was 22.80. (Table 2 and Fig 3) provides the median, minimum and maximum values of Ct values in each disease severity category. Blood samples were serially collected, and number of days after symptom onset was noted, with minimum at 1 day, median of 7 days and maximum at 32 days after symptom onset. Number of days after infection was known in 11 patients and has been included in the analysis.

**Table 1. Analyzes the % inhibition of sVNT to compare between the asymptomatic and symptoms groups, revealing a significant difference with a p-value of less than 0.001.**

|  | Value | df | Asymptotic Significance (2-sided) | Exact Sig. (2-sided) | Exact Sig. (1-sided) |
|---|---|---|---|---|---|
| **Pearson Chi-Square** | 214.737 [a] | 1 | < .001 | < .001 | < .001 |
| **Continuity Correction [b]** | 211.165 | 1 | < .001 | | |
| **Likelihood Ratio** | 232.669 | 1 | < .001 | < .001 | < .001 |
| **Fisher's Exact Test** | | | | < .001 | < .001 |
| **N of Valid Cases** | 395 | | | | |
|  | **Inhibition** | | | | |
| **Mann-Whitney U** | 3132.000 | | | | |
| **Wilcoxon W** | 12862.000 | | | | |
| **Z** | -13.529 | | | | |
| **Asymp. Sig. (2-tailed)** | < .001 | | | | |

a. 0 Cells (0.0%) have expected count less than 5. The minimum expected count is 34.13

b. Computed only for a 2x2 table

a. Grouping Variable: Group

**Table 2. Shows the results of samples from State Quarantine, from persons travelling back from Sudan tested using rRT-PCR, ELISA IgG, IgM and sVNT.**

| | | Interpretation Detected | | |
| --- | --- | --- | --- | --- |
| | | Count | Column N % | Table Total N % |
| Neutralization Ab (sVNT) | N/S | 4 | 20.0% | 2.8% |
| | Negative | 4 | 20.0% | 2.8% |
| | Positive | 12 | 60.0% | 8.3% |
| IgM | Negative | 18 | 90.0% | 12.5% |
| | Positive | 2 | 10.0% | 1.4% |
| Ig | Negative | 4 | 20.0% | 2.8% |
| | Positive | 16 | 80.0% | 11.1% |
| | | Interpretation Invalid internal control | | |
| | | Count | Column N % | Table Total N % |
| **Neutralization Ab (sVNT)** | N/S | 1 | 50.0% | 0.7% |
| | Negative | 1 | 50.0% | 0.7% |
| | Positive | 0 | 0.0% | 0.0% |
| **IgM** | Negative | 2 | 100.0% | 1.4% |
| | Positive | 0 | 0.0% | 0.0% |
| **IgG** | Negative | 2 | 100.0% | 1.4% |
| | Positive | 0 | 0.0% | 0.0% |
| | | Interpretation Not Detected | | |
| | | Count | Column N % | Table Total N % |
| **Neutralization Ab (sVNT)** | N/S | 0 | 0.0% | 0.0% |
| | Negative | 89 | 73.0% | 61.8% |
| | Positive | 33 | 27.0% | 22.9% |
| **IgM** | Negative | 119 | 97.5% | 82.6% |
| | Positive | 3 | 2.5% | 2.1% |
| **IgG** | Negative | 90 | 73.8% | 62.5% |
| | Positive | 32 | 26.2% | 22.2% |

Two hundred fifty-six samples were tested using ELISA IgG, IgM and sVNT, where the results from the three assays were compared (detailed in Table 3). Three samples (1.2%) were negative when tested using sVNT assay, PT042 on day 1 after symptom onset (Ct = 36); PT068 on day 1 after symptom onset (and 4 days after exposure) (Ct = 25.5); PT043 on day 4 after symptom onset (Ct = 35.9). PT002 was positive for sVNT on day 6 and turned negative on day 11. However, both IgM and IgG were positive by these dates. sVNT was evident on sequential samples thereafter in all patients. IgM remained detectable as long as 32 days (last day sample was collected from the patient). Antibodies appeared on the same day of symptom onset in one patient; one day after in 18 and two days after in 19 patients. In two patients, antibodies appeared as early as 4 days after infection (exposure).

Among 77 asymptomatic persons People who tested positive for the COVID virus using the Real-Time PCR method but exhibited no symptoms of the disease at all. in SQ returning from Sudan as representatives of the latter half of 2020, real-time RT-PCR was positive from naso-pharyngeal and throat swab in 15 cases (day 0 = 6, day 3 = 4, day 5 = 4 and day 9 = 1). Results of testing by two RT-PCR protocols were not always agreeable, for example, one of two was positive and not all gene targets designed in each testing could be detected. Twenty-nine (including 11 PCR positive cases) were antibody positive (all sVNT positive with IgG or both IgM and IgG) on day 0 and another remaining 4 PCR positive with antibody negative on day 0 became antibody positive on day 14. The results of this October asymptomatic case study

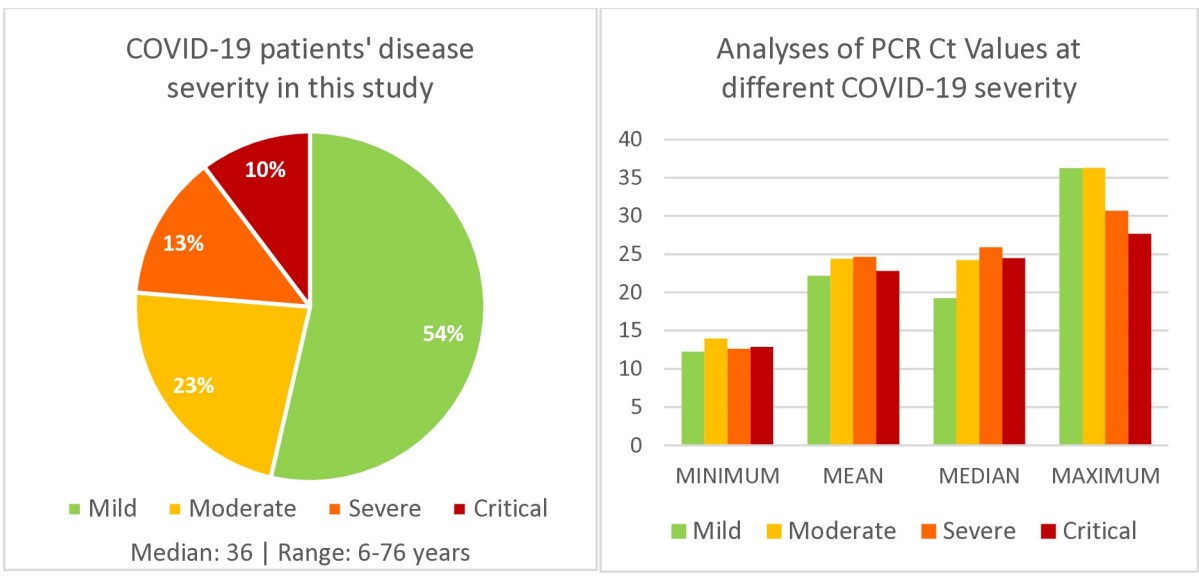

**Fig 3. Analyses of 97 PCR-positive COVID-19 patients against cycle threshold (Ct) values for the different disease severity categories.**
Mild or upper respiratory tract infection; moderate or pneumonia without oxygen desaturation; severe or pneumonia with desaturation, critical or pneumonia requiring mechanical ventilation.

differed from the 2020–2021 symptomatic study in that antibodies were not detectable in all who were PCR positive.

## Conclusion

This study compared the serological test results (ELISA IgG, IgM and sVNT) in 256 samples from 97 PCR-positive COVID-19 symptomatic patients as well as 77 asymptomatic persons in SQ returning from Sudan. The sensitivity of sVNT assay may be compromised in samples collected within 1 and 4 days of symptom onset but compensated by IgM and IgG assay. Unexpectedly, sVNT became negative by day 11, from positive at day 6, after symptom onset in one patient. Hence, all 3 serological parameters should be tested simultaneously to ensure accurate diagnosis. Further, testing IgM alone may be misleading that the infection occurred very recently as IgM was still detectable at day 32 in one patient in this study.

The fact that antibodies assayed in available samples (I point this out since all antibody could show up the same day of symptom onset but we do not have that samples) could be detected very early during the course of the infection, mostly by the first or second day of admission, either on the same day or within one or 2 days after symptom onset, and as early as 4 days after known exposure asserts the importance and reliability of antibody assays as

**Table 3. Analyses the antibodies for the different disease severity categories.**

| | | Disease Severity | | | |
|---|---|---|---|---|---|
| | | Critical Count | Mild Count | Moderate Count | Severe Count |
| IgG | Positive | 32 | 103 | 67 | 54 |
| IgM | Negative | 1 | 0 | 0 | 0 |
| | Positive | 31 | 103 | 67 | 54 |
| Neutralization Ab (sVNT) | Negative | 0 | 3 | 0 | 0 |
| | Positive | 32 | 100 | 67 | 54 |

screening and diagnostic tools in addition to previously advocated use in epidemiological surveillance. Further, NAbs with high percentage inhibition could be found very early regardless of disease severity raises concerns on the value of using therapeutic monoclonal antibodies or convalescent plasma. None of the patients in this study died.

The longest antibody response follow-up in COVID-19 patients in this study was 32 days, which is a limitation of this study. Current published data on how long antibodies is maintained after infection is varied and may be reflection of the condition of the patient, diverse strains of viruses, or the result of influence from therapeutic measures [11].

However, absence of detectable circulating antibody does not negate the possibility of anamnestic response if a COVID-19 patient is re-exposed, if the viruses are not significantly different [13].

In October 2020, testing of asymptomatic persons returning from Sudan trended towards more than false negative real-time RT-PCR results. It remains to be determined whether this trend is due to the evolving status of the virus causing mismatching of primers (data from GISAID). Quality of specimen collection in these samples were assured in the SQ. Thus, to help mitigate false-negatives, SARS-CoV-2 mutations need to be continuously monitored to ensure primers are up to date. Late development of antibody response in the 4 real-time RT-PCR positive cases may not be entirely explained by lack of pro-inflammatory cytokines since all 77 cases remained asymptomatic. Studies of 48 cytokines in many of near-asymptomatic and minimal symptoms infected patients in March-April did not show any significant rise in any [14], yet antibody responses were robust. It is not known whether genomic variations of viruses themselves can cause this variation.

Result of the study in asymptomatic persons there are no symptoms from Sudan suggests there is added value of antibody testing for first day screening of quarantine, and if positive, to perform routine real-time RT-PCR testing. Evaluation on antibody responses using all 3 serological parameters at days 7 or 10 is needed to endorse policy of shortening the quarantine period, which will result in reducing diagnostic cost and SQ burden. Both can help reduce cost burden of SQs while maintaining outbreak vigilance.

Serological testing is an effective method for detecting SARS-CoV-2 infection. It can be utilized in clinical practice to aid in diagnosing infection among groups of patients, whether they exhibit clear symptoms or remain asymptomatic throughout both the initial and later stages of COVID disease. This is particularly valuable in cases where real-time RT-PCR results yield ambiguous or borderline outcomes. Additionally, serological testing helps assess the immune response for protection against SARS-CoV-2 infection in patients.

## Supporting information

**S1 Protocol. ELISA of sera to detect for antibodies against 2019- nCoV protein (IgG).**
(DOCX)

**S2 Protocol. Capture ELISA to detect for IgM antibodies against 2019-nCoV protein.**
(DOCX)

**S1 Fig. Analyses of 97 PCR-positive COVID-19 patients against cycle threshold (Ct) values) for the different disease severity categories.** Mild or upper respiratory tract infection; moderate or pneumonia without oxygen desaturation; severe or pneumonia with desaturation, critical or pneumonia requiring mechanical ventilation.
(TIF)

**S2 Fig. Analyses of 97 PCR-positive COVID-19 patients against % inhibition of sVNT for the different disease severity categories.** Mild or upper respiratory tract infection; moderate

or pneumonia without oxygen desaturation; severe or pneumonia with desaturation, critical or pneumonia requiring mechanical ventilation.
(TIF)

**S3 Fig. Analyses of 97 PCR-positive COVID-19 patients against% sVNT values for the different disease severity categories.** Mild or upper respiratory tract infection; moderate or pneumonia without oxygen desaturation; severe or pneumonia with desaturation, critical or pneumonia requiring mechanical ventilation by One-way ANOVA.
(TIF)

**S1 Table. Shows the results of samples from PCR-positive COVID-19 patients in the first half of 2020 and shows the results of samples from State Quarantine, from persons travelling back from Sudan in the latter half of 2020 tested using rRT-PCR, ELISA IgG, IgM and sVNT.**
(DOCX)

**S2 Table. Shows the results of samples from State Quarantine, from persons travelling back from Sudan in the latter half of 2020 tested using rRT-PCR, ELISA IgG, IgM and sVNT.**
(DOCX)

**S3 Table. Shows the results of analyses of % inhibition of sVNT for the different disease severity categories.**
(DOCX)

## Acknowledgments

This study was supported by Thai Red Cross Emerging Infectious Diseases Health Science Centre, World Health Organization Collaborating Centre for Research and Training on Viral Zoonoses. We thank our colleagues from Institute for Urban Disease Control and Prevention, Department of Disease Control, Ministry of Public Health, Bangkok, Thailand who provided insight and expertise that greatly assisted the study, although they may not agree with all of the conclusions of this study.

We would like to express our sincere gratitude to Prof.Thiravat Haemachudha for his invaluable consultation and guidance on all aspects of this work.

We thank our team for assistance with comments that greatly improved the manuscript.

## Author Contributions

**Data curation:** Thirawat Supharatpariyakorn.

**Formal analysis:** Chanida Ruchisrisarod, Thirawat Supharatpariyakorn.

**Investigation:** Anek Mungaomklang.

**Methodology:** Chanida Ruchisrisarod, Phanni Wanthong, Saowalak Bunprakob.

**Project administration:** Chanida Ruchisrisarod.

**Supervision:** Abhinbhen Saraya Wasontiwong.

**Validation:** Chanida Ruchisrisarod, Yutthana Joyjinda, Pasin Hemachudha, Thirawat Supharatpariyakorn.

**Writing – original draft:** Chanida Ruchisrisarod.

**Writing – review & editing:** Chanida Ruchisrisarod, Pasin Hemachudha, Thiravat Hemachudha.

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
