## [Decision Letter · Decision Letter 0]

21 Nov 2023

PONE-D-23-25209Antibodies response in symptomatic and asymptomatic SARS-CoV-2 infected persons in Thailand.PLOS ONE

Dear Dr. Ruchisrisarod,

Thank you for submitting your manuscript to PLOS ONE. After careful consideration, we feel that it has merit but does not fully meet PLOS ONE’s publication criteria as it currently stands. Therefore, we invite you to submit a revised version of the manuscript that addresses the points raised during the review process.

We look forward to receiving your revised manuscript.

Kind regards,

Yoon-Seok Chung

Academic Editor

PLOS ONE

4. Please include a copy of Table 2 which you refer to in your text on page 8.

Reviewers' comments:

Reviewer's Responses to Questions

**Comments to the Author**

1. Is the manuscript technically sound, and do the data support the conclusions?

Reviewer #1: Partly

Reviewer #2: Yes

2. Has the statistical analysis been performed appropriately and rigorously? 

Reviewer #1: No

Reviewer #2: Yes

3. Have the authors made all data underlying the findings in their manuscript fully available?

Reviewer #1: Yes

Reviewer #2: Yes

4. Is the manuscript presented in an intelligible fashion and written in standard English?

Reviewer #1: Yes

Reviewer #2: Yes

5. Review Comments to the Author

Reviewer #1: This study delineates the result from confirmed patients with the serological test correlated with the clinical classifications. However, this manuscript lacks data on the clinical manifestations in the patient group and all data on the asymptomatic group. Suggests adding all data to make the manuscript complete and cover the aims of this manuscript.

Major concerns.

1. This study involved participants and compared between two groups.

Suggests creating the Demographique table to delineate the characteristics of each group. Te confirmed patients must be presented with the characteristics of clinical manifestations.

Moreover, you can compare between groups by Fisher's exact test, Chi-square test, t-test or Mann-Whitney U test, depending on the outcome type.

2. What are the criteria for diagnosis the asymptomatic patients? Was it by the serological test?

This manuscript did not show any serological test, table or plotted result of this group. Please add it to clarify.

Minor concerns.

1. Figure 2. This figure can subgrouped into four clinical classifications; mild, moderate, severe and critical.

This figure can be presented as an xy-axis with a correlation to see the trend, which could increase the information in this manuscript.

2. Figure 3. This figure must be a pairwise comparison (comparing between groups), not show only the p-value for overall of the overall data set.

Comments.

1. Affiliation. In my understanding, the King Chulalongkorn Memorial Hospital is the largest of the hospitals operated by the Thai Red Cross Society, which is the NGO, not the Government University. This hospital is cooperating with the Chulalongkorn University for medical teaching mission.

Suggests check the first affiliation again.

2. You can be creating the XY plot of the Ct value and sVNT outcomes to see the trend. If you sample both sample types at the same time.

Reviewer #2: Dear Author,

I read your manuscript Antibodies response in symptomatic and asymptomatic SARS-CoV-2 infected persons in Thailand.

As a clinician, evaluation of Antibody assays of IgM, IgG and surrogate isotype independent virus in neutralizing antibody (sVNT) targeting receptor binding domain of Severe Acute Respiratory Syndrome Coronavirus 2 (SARS-CoV-2) in real-time RT-PCR confirmed Coronavirus Disease 2019 (COVID-19) patients with varying severity with a concordance rate of 100% regardless of severity, onset of symptoms and magnitude of viral load is important to shorten length of quarantine among negative cases.

The limitations of the study has already described in the manuscript.

I think that the readers may still benefit from the manuscript.

6. PLOS authors have the option to publish the peer review history of their article (what does this mean?). If published, this will include your full peer review and any attached files.

Reviewer #1: No

Reviewer #2: **Yes: **Elif Gül Yapar Eyi

---

## [Author Response · Author response to Decision Letter 0]

29 Nov 2023

Dear Editor, 

We sincerely thank you for reviewing our manuscript and for the excellent suggestions we received We have made a concerted effort to improve it and agree with all the changes that were made.

 We edit accordingly by reviewers' comments and include statistical analysis information to make it easier to understand. Summarize Table 2 to make it easier to understand and reduce excessive use of data. In this revised version, the sentences are clearer, and the sequence of events and explanations is more organized.

Best regards,

Chanida Ruchisrisarod

Corresponding author

---

## [Decision Letter · Decision Letter 1]

16 Jan 2024

PONE-D-23-25209R1Antibodies response in symptomatic and asymptomatic SARS-CoV-2 infected persons in Thailand.PLOS ONE

Dear Dr. Ruchisrisarod,

Thank you for submitting your manuscript to PLOS ONE. After careful consideration, we feel that it has merit but does not fully meet PLOS ONE’s publication criteria as it currently stands. Therefore, we invite you to submit a revised version of the manuscript that addresses the points raised during the review process.

We look forward to receiving your revised manuscript.

Kind regards,

Yoon-Seok Chung

Academic Editor

PLOS ONE

Journal Requirements:

Reviewers' comments:

Reviewer's Responses to Questions

**Comments to the Author**

1. If the authors have adequately addressed your comments raised in a previous round of review and you feel that this manuscript is now acceptable for publication, you may indicate that here to bypass the “Comments to the Author” section, enter your conflict of interest statement in the “Confidential to Editor” section, and submit your "Accept" recommendation.

Reviewer #1: All comments have been addressed

Reviewer #2: All comments have been addressed

2. Is the manuscript technically sound, and do the data support the conclusions?

Reviewer #1: Partly

Reviewer #2: Partly

3. Has the statistical analysis been performed appropriately and rigorously? 

Reviewer #1: No

Reviewer #2: N/A

4. Have the authors made all data underlying the findings in their manuscript fully available?

Reviewer #1: Yes

Reviewer #2: Yes

5. Is the manuscript presented in an intelligible fashion and written in standard English?

Reviewer #1: Yes

Reviewer #2: No

6. Review Comments to the Author

Reviewer #1: Dear Authors,

I haven't received any response to my previous comments in this manuscript.Please address my comments to improve the manuscript's quality to make sure that it is suitable for publication.Don't forget to enclose a point-by-point response to reviewers and editor.

...

This study delineates the result from confirmed patients with the serological test correlated with the clinical classifications. However, this manuscript lacks data on the clinical manifestations in the patient group and all data on the asymptomatic group. Suggests adding all data to make the manuscript complete and cover the aims of this manuscript.

Major concerns.

1. This study involved participants and compared between two groups.

Suggests creating the Demographique table to delineate the characteristics of each group. Te confirmed patients must be presented with the characteristics of clinical manifestations.

Moreover, you can compare between groups by Fisher's exact test, Chi-square test, t-test or Mann-Whitney U test, depending on the outcome type.

2. What are the criteria for diagnosis the asymptomatic patients? Was it by the serological test?

This manuscript did not show any serological test, table or plotted result of this group. Please add it to clarify.

Minor concerns.

1. Figure 2. This figure can subgrouped into four clinical classifications; mild, moderate, severe and critical.

This figure can be presented as an xy-axis with a correlation to see the trend, which could increase the information in this manuscript.

2. Figure 3. This figure must be a pairwise comparison (comparing between groups), not show only the p-value for overall of the overall data set.

Comments.

1. Affiliation. In my understanding, the King Chulalongkorn Memorial Hospital is the largest of the hospitals operated by the Thai Red Cross Society, which is the NGO, not the Government University. This hospital is cooperating with the Chulalongkorn University for medical teaching mission.

Suggests check the first affiliation again.

2. You can be creating the XY plot of the Ct value and sVNT outcomes to see the trend. If you sample both sample types at the same time.

Reviewer #2: Dear Author,

Thank you for re-submitting. I suggest you to make clear the statement in the Acknowledgments

" We thank our colleagues :from Institute for Urban Disease Control and Prevention, Department of

Disease Control, Ministry of Public Health, Bangkok, Thailand who provided

insight and expertise that greatly assisted the study, although they may not

agree with all of the conclusions of this study." before the editorQs final decision.

7. PLOS authors have the option to publish the peer review history of their article (what does this mean?). If published, this will include your full peer review and any attached files.

Reviewer #1: No

Reviewer #2: **Yes: **Elif Gül YAPAR EYI

---

## [Author Response · Author response to Decision Letter 1]

6 Feb 2024

We sincerely appreciate your thorough review of our manuscript and the valuable suggestions provided. We have diligently worked to enhance the manuscript and have incorporated all the recommended changes. Additionally, we have included statistical analysis information to enhance clarity. To streamline the presentation, we have summarized Table 1 and Table 2 for easier understanding and reduced the excessive use of data. 

We have also added a demographic table to facilitate comparison between each group. Furthermore, we have provided an explanation of the diagnosis of asymptomatic patients. In this revised version, the sentences are clearer, and the sequence of events and explanations is more organized.

---

## [Decision Letter · Decision Letter 2]

23 Apr 2024

PONE-D-23-25209R2Antibodies response in symptomatic and asymptomatic SARS-CoV-2 infected persons in Thailand.PLOS ONE

Dear Dr. Ruchisrisarod,

Thank you for submitting your manuscript to PLOS ONE. After careful consideration, we feel that it has merit but does not fully meet PLOS ONE’s publication criteria as it currently stands. Therefore, we invite you to submit a revised version of the manuscript that addresses the points raised during the review process.

We look forward to receiving your revised manuscript.

Kind regards,

Yoon-Seok Chung

Academic Editor

PLOS ONE

Reviewers' comments:

Reviewer's Responses to Questions

**Comments to the Author**

1. If the authors have adequately addressed your comments raised in a previous round of review and you feel that this manuscript is now acceptable for publication, you may indicate that here to bypass the “Comments to the Author” section, enter your conflict of interest statement in the “Confidential to Editor” section, and submit your "Accept" recommendation.

Reviewer #1: All comments have been addressed

Reviewer #3: (No Response)

Reviewer #4: (No Response)

2. Is the manuscript technically sound, and do the data support the conclusions?

Reviewer #1: Yes

Reviewer #3: No

Reviewer #4: No

3. Has the statistical analysis been performed appropriately and rigorously? 

Reviewer #1: Yes

Reviewer #3: No

Reviewer #4: I Don't Know

4. Have the authors made all data underlying the findings in their manuscript fully available?

Reviewer #1: Yes

Reviewer #3: Yes

Reviewer #4: Yes

5. Is the manuscript presented in an intelligible fashion and written in standard English?

Reviewer #1: Yes

Reviewer #3: No

Reviewer #4: No

6. Review Comments to the Author

Reviewer #1: This revision was fine. However, please carefully align the composition and the aspect ratio (x:y) in Figure 3 to make it readable.

Reviewer #3: Antibodies response in symptomatic and asymptomatic SARS-CoV-2 infected persons in Thailand.

Chanida Ruchisrisarod et al.

General: The paper describes SARS-CoV-2 specific antibody responses in hospital admitted patients or persons in state quarantine. An additional number of serum samples dating before the SARS-CoV-2 pandemic were also included to evaluate test specificity. The results as presented do not provide additional relevant information given the current knowledge on SARS-CoV-2. The results are not sufficiently clearly presented to warrant publication. In addition there are numerous inconsistencies in the paper, some of which I have addressed in the specific comments below.

Specific comments:

Introduction:

Lateral flow tests to detect SARS-CoV-2 antigen are currently widely available and these are much cheaper and easier to apply than the serological tests described in the paper.

“This can be measured by calorimetrically labelling the purified proteins” this should most likely be: This can be measured by enzymatically labelling the purified proteins

Moreover such mutations can affect the diagnostic accuracy of molecular detection such as real-time RT-PCR. The primers usually detect conserved regions in the CoV genome, hence mutations are not expected to be a major issue.

Materials and methods:

The description of the ELISA methos is not clear and the order of some steps is wrong, e.g. substate is added, plates are washed and then stopped. This seems completely wrong. The description of the sVNT is incomplete, i.e. what were the plates coated with?

Statistical analysis:

When are mean ± SD used and when median and IQR? What was done to obtain an underlying normal distribution? The selection of the p value at which to assume statistical significance is not clear.

Results:

Figure 3. What are the errorbars? There is an ANOVA p value in the figure, but eyeballing the data suggests that these are not-normally distributed.

Reviewer #4: Poor English in many situations

SARS-CoV-2 antibody detection:

ELISA in-house kit testing for IgG, IgM and sVNT were developed:

Validation of in-house kit should be done before use in real tests. In text this not sated.

Sources of materials are not indicated.

Preparation of ELISA plate and running of the test are merged. Preparation and validation of the in-house prepared kit should be done first and then being used to test sera.

Please verify in-house kit and then us in test separately.

Also, IgM detection is badly written.

“Fifty μl of TMB substrate was added to each well and incubated in a dark room at room temperature. Wells were washed five times with PBST 300μl per well, after which the reaction was stopped using sulphuric acid.” This absolutely cannot be done in ELISA technique.

Validation of the in-house kits is indicated at the end of the method section.

7. PLOS authors have the option to publish the peer review history of their article (what does this mean?). If published, this will include your full peer review and any attached files.

Reviewer #1: No

Reviewer #3: No

Reviewer #4: No

---

## [Author Response · Author response to Decision Letter 2]

8 May 2024

Dear Reviewers, 

We sincerely appreciate your comprehensive review of our manuscript and the valuable suggestions provided. We have diligently revised the manuscript to incorporate all the recommended changes and enhancements. Additionally, we have included statistical analysis information to improve comprehension, expanded the information in the introduction section, and revised the experimental method to enhance clarity, as per the reviewer's recommendations.

Additionally, we have incorporated a demographic table to facilitate comparisons between each group. Furthermore, we have included an explanation of the diagnosis of asymptomatic patients. In this revised version, the sentences have been refined for clarity, and the sequence of events and explanations has been reorganized for improved coherence.

Best regards,

Chanida Ruchisrisarod

Corresponding author

---

## [Decision Letter · Decision Letter 3]

1 Aug 2024

Antibodies response in symptomatic and asymptomatic SARS-CoV-2 infected persons in Thailand.

PONE-D-23-25209R3

Dear Dr. Ruchisrisarod,

We’re pleased to inform you that your manuscript has been judged scientifically suitable for publication and will be formally accepted for publication once it meets all outstanding technical requirements.

Kind regards,

Vittorio Sambri, M.D., Ph.D.

Academic Editor

PLOS ONE

Additional Editor Comments (optional):

Reviewers' comments:

Reviewer's Responses to Questions

**Comments to the Author**

1. If the authors have adequately addressed your comments raised in a previous round of review and you feel that this manuscript is now acceptable for publication, you may indicate that here to bypass the “Comments to the Author” section, enter your conflict of interest statement in the “Confidential to Editor” section, and submit your "Accept" recommendation.

Reviewer #1: All comments have been addressed

Reviewer #5: All comments have been addressed

2. Is the manuscript technically sound, and do the data support the conclusions?

Reviewer #1: Yes

Reviewer #5: Yes

3. Has the statistical analysis been performed appropriately and rigorously? 

Reviewer #1: Yes

Reviewer #5: Yes

4. Have the authors made all data underlying the findings in their manuscript fully available?

Reviewer #1: Yes

Reviewer #5: Yes

5. Is the manuscript presented in an intelligible fashion and written in standard English?

Reviewer #1: Yes

Reviewer #5: Yes

6. Review Comments to the Author

Reviewer #1: Comments.

1. Figure 3: Suggest deleting "120" in the y-axis. The theoretically of "%inhibition" must not over 100% because of the measurement.

Reviewer #5: (No Response)

7. PLOS authors have the option to publish the peer review history of their article (what does this mean?). If published, this will include your full peer review and any attached files.

Reviewer #1: No

Reviewer #5: No

---

## [Editor Report · Acceptance letter]

21 Jan 2025

PONE-D-23-25209R3 

PLOS ONE

Dear Dr. Ruchisrisarod, 

I'm pleased to inform you that your manuscript has been deemed suitable for publication in PLOS ONE. Congratulations! Your manuscript is now being handed over to our production team.

Kind regards, 

on behalf of

Professor Vittorio Sambri 

Academic Editor

PLOS ONE